# FuseRL: Dense Preference Optimization for Heterogeneous Model Fusion

## Abstract

Heterogeneous model fusion enhances the performance of LLMs by integrating the knowledge and capabilities of multiple structurally diverse models. However, existing approaches often rely solely on selecting the best output for each prompt from source models, which underutilizes their full potential due to limited source knowledge and results in sparse optimization signals. To address this limitation, we propose FuseRL, a novel two-stage framework comprising FuseSFT and FusePO to maximize the utilization of source LLMs. FuseSFT establishes a robust initialization by integrating the strengths of heterogeneous source models through weighted supervised fine-tuning (SFT) on diverse outputs for each prompt. FusePO optimizes weighted preferences based on the outputs of multiple source models to enable superior alignment performance. Extensive experiments demonstrate the effectiveness of our framework across various preference alignment methods, including RLOO, DPO, and SimPO. Using Llama-3.1-8B-Instruct as the target model, our approach achieves competitive performance among 8B LLMs on the AlpacaEval-2 and Arena-Hard benchmarks. Further analysis suggests that FuseSFT regularizes the training to reduce overfitting, while FusePO introduces dense and diverse preference signals that enhance alignment quality.

## 1 Introduction

Leveraging the collective knowledge and unique strengths of multiple large language models (LLMs) presents a highly promising avenue for enhancing generalization, robustness, and efficiency across a wide range of complex and diverse tasks. The underlying rationale is that no single LLM—particularly when constrained by scale or data—can comprehensively capture the full spectrum of task complexity and domain variability. Representative strategies to achieve this objective include ensemble methods (Aniol et al., 2019; Jiang et al., 2023b; Xu et al., 2024), Mixture of Experts (MoE) (Fedus et al., 2022; Sukhbaatar et al., 2024), model merging (Wortsman et al., 2022; Akiba et al., 2024), and heterogeneous model fusion (Wan et al., 2024a;b; Shi et al., 2024; Yang et al., 2024c). While these techniques share the common goal of integrating multiple LLMs to capitalize on their collective strengths, each comes with its own advantages and challenges.

Ensemble methods combine the outputs of multiple models to generate more robust predictions. However, they typically require running all constituent models simultaneously, resulting in substantial memory and computational overhead. MoE partially alleviates these efficiency challenges by activating only a subset of parameters during inference. Nonetheless, the entire model generally remains loaded in memory, and training MoE systems can be resource-intensive. Model merging integrates models with identical architectures into a unified parameter set, enhancing robustness and generalization but limiting applicability to homogeneous model families. In contrast, heterogeneous model fusion employs techniques like multi-teacher knowledge distillation to transfer complementary expertise across diverse model configurations. However, these methods often require complex vocabulary alignment to fuse the output distributions of component models. Implicit model fusion (IMF) addresses this challenge by directly utilizing the outputs (responses) of source models for heterogeneous model fusion. For example, WRPO (Yang et al., 2024c) employs progressive adaptation to gradually shift optimization from target model outputs to high-quality source model responses.

Moreover, existing heterogeneous model fusion methods often face another critical challenge: they limit their potential by focusing exclusively on selecting the best output for each prompt from source

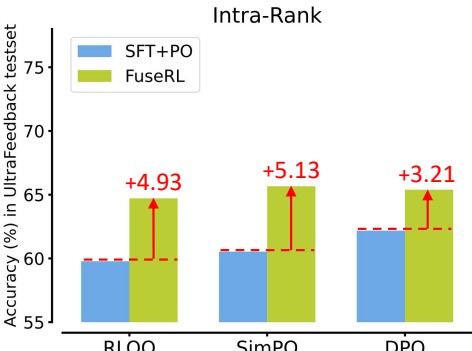 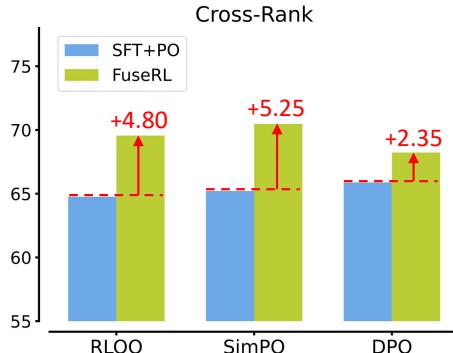

Figure 1: Effect of using a single (SFT+PO) vs. multiple (FuseRL) source LLMs for each prompt for heterogeneous model fusion on UltraFeedback (Cui et al., 2024). *Accuracy* (Meng et al., 2024) measures the ability to accurately distinguish between preferred and dispreferred responses by comparing the average log-probabilities assigned by different fused models. **Left:** Accuracy for multiple responses generated from a single source model. **Right:** Accuracy across responses generated by different source models. Compared to directly applying SFT followed by preference optimization (SFT+PO), FuseRL demonstrates superior performance in distinguishing responses, which reflects improved alignment with human preferences. More details are provided in Appendix D.1.

models. This narrow and static reliance on source knowledge introduces notable drawbacks, primarily stemming from bias and limited response diversity. The preferences generated by a single model reflect its unique strengths, weaknesses, and inherent response distribution, which can introduce systematic errors and restrict the variety of training data. This may result in a biased policy that overfits to the specific characteristics of that model and struggles to generalize to broader scenarios. Furthermore, the lack of diversity in source model responses limits the policy's ability to learn from a wide range of high-quality examples, leading to sparse training signals.

This paper focuses on improving the utilization of source LLMs and providing denser, more diverse training signals for implicit model fusion. To this end, we introduce **FuseRL**, a novel reinforcement learning framework specifically designed to unlock the potential of fusing diverse source models through a two-stage process. **FuseSFT**: This stage improves the target model by fine-tuning it with high-quality responses from multiple source models. By employing a reward-based mechanism, FuseSFT prioritizes responses with high informativeness and relevance and establishes a strong foundation for subsequent fusion training. Moreover, FuseSFT effectively mitigates the *squeezing effect* (Ren & Sutherland, 2025), which can emerge during SFT and impede subsequent preference optimization. **FusePO**: Building upon the initialization from FuseSFT, FusePO aligns the target model with human preferences by dynamically leveraging weighted preference signals derived from multiple source models. This stage emphasizes high-reward preferences while maintaining adaptability across various preference optimization methods. By improving the integration of heterogeneous capabilities and maximizing the utilization of source model outputs, our framework aims to provide a more robust approach to heterogeneous model fusion. In Figure 1, we present a preliminary experiment exploring how FuseRL impacts the model's ability to distinguish response quality. The results demonstrate that more effective utilization of diverse source models leads to richer and denser preference signals and improved alignment with human preferences.

Extensive experiments validate the effectiveness of our framework across various preference alignment methods, including RLOO, DPO, and SimPO. Our approach achieves state-of-the-art performance among 8B-sized LLMs on the AlpacaEval-2 and Arena-Hard benchmarks. Further analysis shows that fully leveraging the responses from multiple LLMs mitigates the bias introduced when relying on a single model, resulting in more diverse preference signals that better approximate the true reward distribution. Moreover, weighting preferences by their associated rewards reduces variance in the training signals by prioritizing high-quality, informative samples and down-weighting suboptimal ones. By reducing both bias and variance, the policy is able to learn from diverse data and dense signals, which in turn improves generalization and ensures stable and efficient convergence.

## 2 PRELIMINARIES

Reinforcement learning from human feedback (RLHF) (Christiano et al., 2017) is a framework for aligning LLMs with human preferences. The training objective in RLHF is to optimize a policy $\pi_\theta$ to maximize reward signals from human feedback while constraining excessive deviations from a reference policy $\pi_{\text{ref}}$:

$$J(\pi_\theta) = \mathbb{E}_{x \sim \mathcal{D}, y \sim \pi_\theta} \big[ r(x, y) \big] - \beta \, \text{KL}(\pi_\theta \| \pi_{\text{ref}}), \tag{1}$$

where $r(x, y)$ is a reward function that captures human preferences for a prompt $x$ and response $y$, $\text{KL}(\pi_\theta \| \pi_{\text{ref}})$ penalizes deviations of the policy $\pi_\theta$ from the reference policy $\pi_{\text{ref}}$, and $\beta$ controls the trade-off between maximizing the overall reward and maintaining adherence to the reference policy. This trade-off ensures stability during training and mitigates risks such as mode collapse.

**REINFORCE** The REINFORCE (Williams, 1992) algorithm is a classic policy gradient method that can be adapted to implement the RLHF objective. REINFORCE updates the policy by maximizing the expected reward through gradient ascent. The policy gradient is given by:

$$\nabla_\theta J(\pi_\theta) = \mathbb{E}_{x \sim \mathcal{D}, y \sim \pi_\theta} \left[ \nabla_\theta \log \pi_\theta(y|x) \cdot \hat{r}(x, y) \right], \tag{2}$$

where $\hat{r}(x, y) = r(x, y) - \beta \, \nabla_\theta \text{KL}(\pi_\theta(\cdot|x) \| \pi_{\text{ref}}(\cdot|x))$ is the adjusted reward with KL penalty.

To further stabilize training, a baseline $b$ can be introduced into the objective function of REINFORCE to reduce the variance of reward estimates while maintaining their unbiased nature. REINFORCE Leave-One-Out (RLOO) (Kool et al., 2019) estimates the baseline $b$ using multiple online samples: $b(x, y_i) = \frac{1}{k-1} \sum_{j \neq i} \hat{r}(x, y_j)$, where $y_i$ represents the $i$th response independently sampled from the policy $\pi_\theta$ conditioned on the prompt $x$. With the baseline term, the adjusted reward in Eq. (2) becomes:

$$\hat{r}(x, y) = r(x, y) - \beta \, \nabla_\theta \text{KL}(\pi_\theta(\cdot|x) \| \pi_{\text{ref}}(\cdot|x)) - b(x, y).$$

**Direct Preference Optimization (DPO)** DPO is an offline preference optimization method that directly aligns LLMs with human preferences, offering an alternative to traditional RLHF. Unlike RLHF, which relies on reinforcement learning to optimize a reward model and iteratively improve the policy, DPO builds on the Bradley-Terry (BT) objective (Bradley & Terry, 1952). This objective models the probability of the preferred response $y_w$ being ranked higher than the dispreferred response $y_l$:

$$p(y_w \succ y_l | x) = \sigma(r(x, y_w) - r(x, y_l)), \tag{3}$$

where $r(x, y)$ is the reward function, and $\sigma$ is the sigmoid function. DPO reparameterizes $r(x, y)$ in Eq. (1) as:

$$r(x, y) = \beta \log \frac{\pi_\theta(y|x)}{\pi_{\text{ref}}(y|x)} + \beta \log Z(x), \tag{4}$$

where $Z(x) = \sum_y \pi_{\text{ref}}(y|x) \exp\big(\frac{1}{\beta} r(x, y)\big)$ is the partition term. From this formulation, DPO defines its objective as:

$$\mathcal{L}_{\text{DPO}}(\pi_\theta; \pi_{\text{ref}}) = -\mathbb{E}_{(x, y_w, y_l) \sim \mathcal{D}}[\log p(y_w \succ y_l | x)]. \tag{5}$$

SimPO (Meng et al., 2024) extends DPO by introducing a reference-free reward formulation:

$$r_{\text{SimPO}}(x, y) = \frac{\beta}{|y|} \log \pi_\theta(y|x). \tag{6}$$

To enhance the differentiation between preferred and non-preferred responses, SimPO further introduces a reward margin $\gamma$ and modifies the BT probability as:

$$p(y_w \succ y_l | x) = \sigma(r_{\text{SimPO}}(x, y_w) - r_{\text{SimPO}}(x, y_l) - \gamma). \tag{7}$$

## 3 METHODOLOGY

To enhance the utilization of outputs from multiple source models for implicit model fusion, we propose a novel two-stage framework, FuseRL, which consists of two key components: FuseSFT and FusePO. FuseSFT fine-tunes the target model using high-quality responses from multiple source models, prioritizing those with greater informativeness and relevance. FusePO further aligns the target model with human preferences by leveraging weighted preference signals, emphasizing high-reward responses while ensuring robustness and applicability across various preference optimization methods. An overview of this framework is illustrated in Figure 2.

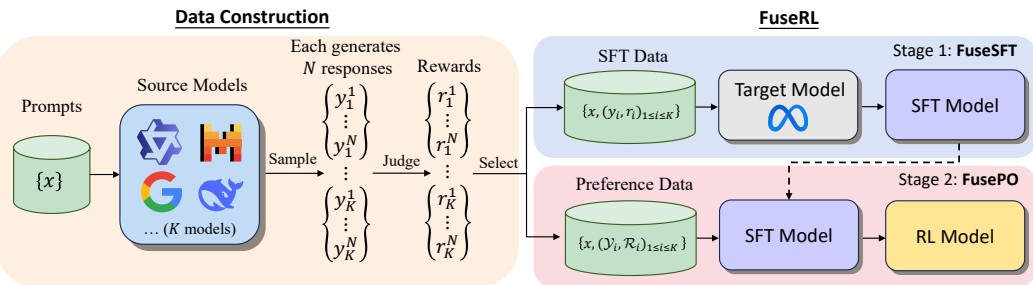

Figure 2: Overview of the proposed FuseRL framework. It comprises two stages: FuseSFT, which fine-tunes the target model using high-quality responses from diverse source models via a reward-based mechanism to prioritize informative and relevant outputs; and FusePO, which dynamically adjusts weighted preference pair contributions to align the target model with human preferences.

### 3.1 NOTATIONS

Our approach begins by constructing data samples capturing strengths of multiple source models. This ensures the target model is trained on diverse and informative responses.

Given $K$ source models $\mathcal{M} = \{M_1, M_2, \ldots, M_K\}$, each $M_i$ generates a response set $\mathcal{Y}_i$ for a given input $x \in \mathcal{X}$: $\mathcal{Y}_i = \{y_i^1, y_i^2, \ldots, y_i^N\}$, for $i = 1, 2, \ldots, K$. An external reward model is then used to assign a reward score $r(x, y)$ to each response $y \in \mathcal{Y}_i$, resulting in the reward set $\mathcal{R}_i = \{r(x, y_i^1), r(x, y_i^2), \ldots, r(x, y_i^N)\}$.

To regulate the contributions of the source models, we assign a weight to each model for a given input $x$. Let $y_i = \arg\max_{y \in \mathcal{Y}_i} r(x, y)$ represent the response from source model $M_i$ that achieves the highest reward given $x$. The weight for model $M_i$ is defined as:

---

**Algorithm 1 Data Construction and Weighting**

**INPUT:** Instruction set $\mathcal{X}$, source models $\mathcal{M} = \{M_1, M_2, \ldots, M_K\}$, reward model $r(x, y)$.
**Data Split:** Split the instruction set into:
$$\mathcal{X} = \mathcal{X}_{\text{sft}} \cup \mathcal{X}_{\text{po}}, \quad \mathcal{X}_{\text{sft}} \cap \mathcal{X}_{\text{po}} = \varnothing.$$
**Sampling and Weighting:**
  **for** each $x$ in $\mathcal{X}_{\text{sft}}$ or $\mathcal{X}_{\text{po}}$ **do**
    **for** each $M_i \in \mathcal{M}$ **do**
      Generate responses: $\mathcal{Y}_i = \{y_i^1, \ldots, y_i^N\}$.
      Compute rewards: $\mathcal{R}_i = \{r(x, y_i^j)\}_{j=1}^N$.
      Select $y_i = \arg\max_{y \in \mathcal{Y}_i} r(x, y)$.
    **end for**
    Compute weight: $w_{x,i} = \frac{\exp(r(x,y_i)/\alpha)}{\sum_{i=1}^K \exp(r(x,y_i)/\alpha)}$

    Store $\{x, y_i, \mathcal{Y}_i, \mathcal{R}_i, w_{x,i}\}$ for each $M_i$.
  **end for**

---

$$w_{x,i} = \frac{\exp\left(\frac{r(x,y_i)}{\alpha}\right)}{\sum_{i=1}^K \exp\left(\frac{r(x,y_i)}{\alpha}\right)}, \tag{8}$$

where $\alpha$ is the temperature coefficient. Refer to Algorithm 1 for further details.

### 3.2 FUSESFT

Given a prompt $x$ and response $y$, the supervised fine-tuning (SFT) objective for the target model $\pi_{\theta_T}$ is defined as:

$$\mathcal{L}_{\text{SFT}}(y, x; \pi_{\theta_T}) = -\log \pi_{\theta_T}(y|x). \tag{9}$$

FuseSFT extends the standard SFT objective by utilizing responses $\{y_i\}_{i=1}^K$ generated from all $K$ source models to prioritize those with higher informativeness and relevance, which establishes a robust foundation for subsequent optimization. Using a similar weighting scheme as defined in Eq. (8), FuseSFT applies a weighted combination of the highest-reward responses during fine-tuning: [1]

$$\mathcal{L}_{\text{FuseSFT}} = \sum_{x \in \mathcal{X}} \sum_{i=1}^K w_{x,i} \cdot \mathcal{L}_{\text{SFT}}(y, x; \pi_{\theta_T}). \tag{10}$$

While FuseSFT is designed to leverage high-quality responses from multiple source models, its benefits extend beyond simple data aggregation. Recent work on learning dynamics in LLM finetuning

---

[1] The rationale for FuseSFT's modified weighting scheme is discussed in Section 4.1 and Appendix J.

(Ren & Sutherland, 2025) reveals that the early-stage supervision signal plays a critical role in shaping the model's future behavior—especially in downstream preference optimization. Overly confident or homogeneous supervision can lead to a compression of gradient signals (the *squeezing effect*), which may adversely affect the model's alignment performance. By incorporating diverse responses and weighting them softly rather than selecting only the highest-scoring one, FuseSFT mitigates this effect and better preserves the gradient diversity necessary for effective preference learning.

### 3.3 FUSEPO

Building on FuseSFT, FusePO aims to dynamically leverage weighted preference signals derived from multiple source models. By prioritizing high-reward preferences, it optimizes the target model using diverse and high-quality preference pairs. Moreover, FusePO employs a general preference learning loss function, $\mathcal{L}_{\text{pref}}$, which can be instantiated with methods such as

---

**Algorithm 2 DPO-Implemented FuseRL**

---

**INPUT:** Target model $\pi_{\theta_T}$, learning rates $\eta_{\text{sft}}$ and $\eta_{\text{po}}$, constructed data from Algorithm 1.

**STAGE 1: FuseSFT**

  **for** each prompt $x$ in $\mathcal{X}_{\text{sft}}$ **do**

    **for** each source model $M_i \in \mathcal{M}$ **do**

      Retrieve $\{x, y_i, \mathcal{Y}_i, \mathcal{R}_i, w_{x,i}\}$.

    **end for**

    $\theta_T \leftarrow \theta_T - \eta_{\text{sft}} \cdot \nabla_{\theta_T} \mathcal{L}_{\text{FuseSFT}}$.

  **end for**

**STAGE 2: FusePO**

  **for** each prompt $x$ in $\mathcal{X}_{\text{po}}$ **do**

    **for** each source model $M_i \in \mathcal{M}$ **do**

      Retrieve $\{x, y_i, \mathcal{Y}_i, \mathcal{R}_i, w_{x,i}\}$.

      Form preference data $(x, y_i^w, y_i^l)$ from $\mathcal{Y}_i$ and $\mathcal{R}_i$.

    **end for**

    Minimize Eq. (11): $\theta_T \leftarrow \theta_T - \eta_{\text{po}} \cdot \nabla_{\theta_T} \mathcal{L}_{\text{FusePO}}$.

  **end for**

**OUTPUT:** Final fused model $\theta_T^* \leftarrow \theta_T$.

---

RLOO, DPO, or others. Unlike FuseSFT, FusePO leverages the complete response set $\mathcal{Y}_i$ from each source model to construct training data. For instance, responses from the same source model are used to create preference pairs when necessary to minimize distributional variance and enhance the overall learning process. Specifically, the FusePO loss function is defined as:

$$\mathcal{L}_{\text{FusePO}} = \sum_{x \in \mathcal{X}} \sum_{i=1}^{K} w_{x,i} \cdot \mathcal{L}_{\text{pref}}(\mathcal{Y}_i, \mathcal{R}_i, x; \pi_{\theta_T}). \tag{11}$$

In this work, we investigate the implementation of $\mathcal{L}_{\text{pref}}$ using various preference optimization methods, including RLOO, DPO, and SimPO (see experiments). Furthermore, to better illustrate FuseRL, we use DPO as an example to outline the process in Algorithm 2.

## 4 EXPERIMENTS

### 4.1 EXPERIMENTAL SETUPS

**Models for fusion.** In our experiments, we utilize four diverse open-source LLMs as source models: Mistral-Large-Instruct-2407 (Jiang et al., 2023a), Gemma2-27B-IT (Riviere et al., 2024), Qwen2.5-72B-Instruct (Yang et al., 2024b), and DeepSeek-V2-Chat-0628 (Shao et al., 2024). These models were chosen for their diverse architectures, parameter scales, and complementary strengths, aligning with our goal of heterogeneous model fusion. For the target model, we employ Llama-3.1-8B-Instruct (Dubey et al., 2024) for its balance of efficiency and performance.

**Preference optimization methods.** To assess the generalizability of our FuseRL framework, we implement RLOO (Ahmadian et al., 2024), DPO (Rafailov et al., 2023), and SimPO (Meng et al., 2024) in the main experiments. RLOO serves as a traditional reinforcement learning algorithm, whereas DPO and SimPO represent reference-based and reference-free preference optimization methods, respectively. The notable distinctions among these algorithms offer a solid foundation for evaluating the adaptability of our framework.

**Baselines.** We evaluate our method with various baseline models, including proprietary LLMs, source and target LLMs, ensemble LLMs, and prior approaches for heterogeneous model fusion. Due to space limitations, detailed descriptions of all baseline settings are provided in Appendix D.2.

**Training dataset.** We utilize UltraFeedback (Cui et al., 2024) as our training dataset. Ultra-Feedback is a large-scale preference dataset containing approximately 64,000 samples, primarily

Table 1: Results of FuseRL and baselines on AlpacaEval-2 and Arena-Hard. All methods are evaluated using GPT-4-1106-Preview as the judge model. **Bolded** numbers indicate the best performance and underlined numbers suggest the second-best performance. Scores in parentheses indicate the points of increase or decrease relative to the counterpart in the previous row.

| Model | Size | AlpacaEval-2 | | | Arena-Hard | | |
|---|---|---|---|---|---|---|---|
| | | LC (%) | WR (%) | Avg. Len. | SC (%) | WR (%) | Avg. Len. |
| *Proprietary LLMs* | | | | | | | |
| GPT-4o | - | 57.5 | 51.3 | 1873 | 69.9 | 79.2 | 2988 |
| GPT-4-Turbo | - | 50.0 | 50.0 | 2049 | 50.0 | 50.0 | 2748 |
| *Source&Target LLMs* | | | | | | | |
| Llama-3.1-8B-Instruct | 8B | 28.3 | 28.7 | 1962 | 23.8 | 28.1 | 2695 |
| Mistral-Large-Instruct | 123B | 54.3 | 46.8 | 1771 | 63.1 | 70.4 | 1762 |
| Gemma2-27B-IT | 27B | 55.5 | 41.0 | 1558 | 47.4 | 57.5 | 2545 |
| Qwen2.5-72B-Instruct | 72B | 50.9 | 55.2 | 2249 | 63.4 | 78.0 | 3446 |
| DeepSeek-V2-Chat | 236B | 45.9 | 40.7 | 1843 | 58.9 | 68.6 | 2732 |
| *Ensemble LLMs* | | | | | | | |
| GPT4-Top1 | 458B | 72.1 | 72.0 | 2171 | 92.2 | 94.9 | 3157 |
| LLM-Blender-Top1 | 458B | 55.6 | 49.7 | 1857 | 55.9 | 66.2 | 2675 |
| MoA | 458B | 58.7 | 76.8 | 2982 | 72.7 | 87.1 | 4243 |
| *Heterogeneous Model Fusion* | | | | | | | |
| FuseLLM | 8B | 36.0 | 33.8 | 1930 | 24.6 | 32.1 | 2585 |
| FuseChat | 8B | 38.1 | 35.2 | 1866 | 24.8 | 32.7 | 2653 |
| WRPO | 8B | 67.7 | **74.1** | 2493 | 40.5 | 58.1 | 3801 |
| SFT | 8B | 41.5 | 38.6 | 1901 | 28.8 | 40.2 | 2831 |
| FuseSFT | 8B | 38.8 (-2.7) | 33.7 (-4.9) | 1805 | 26.4 (-2.4) | 35.8 (-4.4) | 2672 |
| SFT + RLOO | 8B | 59.0 | 63.3 | 2315 | 36.5 | 53.4 | 3324 |
| FuseRL$_{RLOO}$ (Ours) | 8B | 67.7 (+8.7) | 70.6 (+7.3) | 2324 | 40.8 (+4.3) | **58.6** (+5.2) | 3523 |
| SFT + SimPO | 8B | 64.7 | 67.6 | 2269 | 39.8 | 55.6 | 3343 |
| FuseRL$_{SimPO}$ (Ours) | 8B | **70.6** (+5.9) | 71.3 (+3.7) | 2172 | 41.2 (+1.4) | 56.4 (+0.8) | 2866 |
| SFT + DPO | 8B | 67.1 | 69.8 | 2249 | 42.2 | 57.6 | 3360 |
| FuseRL$_{DPO}$ (Ours) | 8B | 70.1 (+3.0) | 70.9 (+1.1) | 2152 | **43.7** (+1.5) | 57.5 (-0.1) | 3060 |

focused on areas such as instruction-following, truthfulness, honesty, and helpfulness. To implement FuseRL, we sample responses from various source models for each prompt in the dataset. Specifically, each source model generates five responses per prompt using top-$p$ sampling (see Appendix D.3 for sampling details). These responses are then evaluated by an external reward model, ArmoRM-Llama3-8B-v0.1 (Wang et al., 2024a).

We partition the dataset into two splits with a 4:6 ratio for our two-stage training process. In the FuseSFT stage, we aggregate responses generated by the source models and select the top four responses based on reward scores. This strategy balances diversity with the quality of training samples. As shown in our comparative analysis in Appendix J, this selection method outperforms selecting the best responses solely from individual source models. In the FusePO stage, due to computational constraints, we select two responses per source model: one with the highest reward score and one with the lowest RM score among the five sampled responses, forming $\mathcal{Y}_i$. Detailed training hyperparameters and implementation specifics are provided in Appendix D.3.

**Evaluation.** We assess the performance of our model on two widely recognized evaluation benchmarks in the research community: AlpacaEval-2 (Li et al., 2023; Dubois et al., 2024) and Arena-Hard (Li et al., 2024). AlpacaEval-2 comprises 805 questions sourced from five diverse datasets. We evaluate performance using two metrics: length-controlled (LC) win rate and raw win rate (WR), benchmarking against GPT-4-Preview-1106. The judge model for this evaluation is also GPT-4-Preview-1106. Arena-Hard consists of 500 challenging user queries derived from Chatbot Arena (Chiang et al., 2024), with performance metrics including style-controlled (SC) win rate and raw win rate (WR), compared against GPT-4-0314. The judge model employed for Arena-Hard evaluation is GPT-4-Preview-1106. These benchmarks were selected for their capacity to comprehensively evaluate the model's conversational capabilities. Furthermore, we present the performance of FuseRL across a broader range of downstream tasks, including question answering, reasoning, mathematics, and coding. Due to space limitations, detailed results are provided in Appendix F.

## 4.2 OVERALL RESULTS

Table 1 presents the results of our method compared to a range of strong baseline methods on both AlpacaEval-2 and Arena-Hard. Based on the experimental results, we identify several key insights.

Firstly, through our two-stage training process, FuseRL achieves substantial performance gains compared to the initial Llama-3.1-8B-Instruct (target model) on both AlpacaEval-2 and Arena-Hard benchmarks. Specifically, FuseRL$_{DPO}$ demonstrates an impressive 41.8-point improvement in LC win rate on AlpacaEval-2 and a 19.9-point improvement in SC win rate on Arena-Hard. Moreover, FuseRL outperforms all source LLMs and proprietary LLMs on AlpacaEval-2, including Qwen-2.5-72B-Instruct, Mistral-Large-Instruct, GPT-4-Turbo, and GPT-4o, and others.

Secondly, in comparison to ensemble LLM methods, FuseRL achieves higher LC win rates than both LLM-Blender-Top1 and MoA on the AlpacaEval-2 benchmark. Notably, considering that GPT4-Top1 represents a strong and surposable upper bound for fusion performance (Wan et al., 2024b; Yang et al., 2024c), it is remarkable that FuseRL closely approximates this upper bound on AlpacaEval-2, despite being much smaller in size. However, the performance gap with GPT4-Top1 on Arena-Hard is significantly larger. We argue that this performance discrepancy stems from inherent differences in the distribution and complexity of prompts between UltraFeedback and Arena-Hard, as visually illustrated in Figure 3.[2]

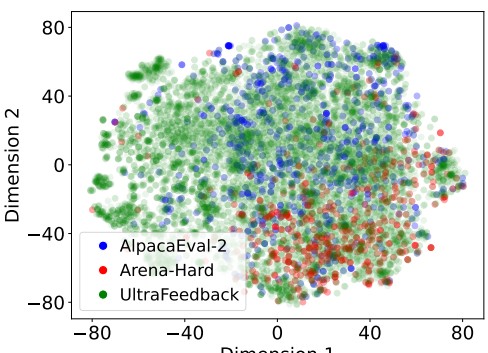

Figure 3: t-SNE visualization of prompts from UltraFeedback, AlpacaEval-2, and Arena-Hard. The prompt embeddings are projected via t-SNE. While AlpacaEval-2 prompts are distributed relatively evenly across the UltraFeedback distribution, the Arena-Hard prompts show a more pronounced distributional deviation.

Thirdly, our proposed FuseRL consistently outperforms previous heterogeneous model fusion techniques, including FuseLLM, FuseChat, and WRPO. Specifically, compared to the most relevant baseline, WRPO, our FuseRL$_{DPO}$ achieves improvements of 2.4 points on AlpacaEval-2 and 3.2 points on Arena-Hard. Furthermore, when compared to using only the best individual source model for each prompt (i.e., SFT+RLOO, SFT+DPO, or SFT+SimPO), FuseRL delivers substantial gains across all configurations—RLOO, DPO, and SimPO. Notably, the performance of RLOO is comparatively lower than that of DPO and SimPO, likely due to the limited number (two) of responses used for each prompt, which constrains its overall performance. These results collectively underscore the effectiveness of FuseRL in leveraging the dense and diverse preference signals from heterogeneous source models to drive more robust and superior alignment performance.

### 4.3 FUSESFT AND FUSEPO: ABLATION STUDIES

Table 1 reveals another intriguing phenomenon: while the target model trained solely with SFT initially outperforms the FuseSFT model, the FuseSFT model achieves superior performance after the second stage. Furthermore, as illustrated in Figure 4, the target model consistently shows improved performance after applying FuseSFT across all (off-policy) preference optimization methods, including DPO, SimPO, and RLOO. This observation indicates that the alignment performance achieved during the first stage does not necessarily determine the eventual performance gains realized through subsequent preference learning. Although FuseSFT may not yield better alignment results in the first stage, it enhances the effectiveness of preference learning from source models in the second stage. We speculate that this is due to two primary factors. First, learning from multiple responses, rather than focusing solely on the highest-scoring response, introduces additional

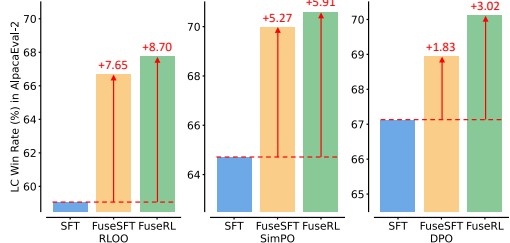

Figure 4: Ablation studies for FuseRL across various preference learning methods, including RLOO, SimPO, and DPO. SFT refers to applying standard supervised fine-tuning on the target model, while FuseSFT extends this by incorporating multiple responses from source models. FuseRL combines FuseSFT and FusePO.

---

[2] We use all-mpnet-base-v2 from `https://huggingface.co/sentence-transformers/all-mpnet-base-v2` to generate prompt embeddings.

Table 2: AlpacaEval-2 results of FuseRL with varying numbers of source models (1, 2, and 4), where the number of models is the only varying factor and all are used consistently in both the FuseSFT and FusePO stages. The results show that increasing the number of source models during fusion leads to consistent improvements in both the LC win rate and WR.

| Method | # Source Models | AlpacaEval-2 | |
|---|---|---|---|
| | | LC (%) | WR (%) |
| FuseRL$_{DPO}$ | 4 | **70.1** | **70.9** |
| | 2 | 69.1 | 66.8 |
| | 1 | 65.5 | 62.4 |

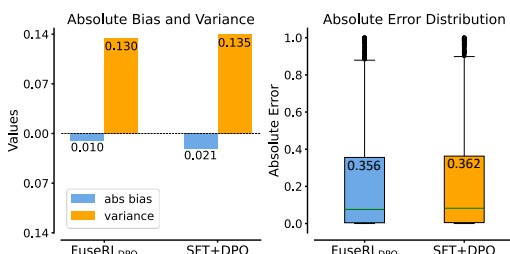

Figure 5: Results of FuseRL$_{DPO}$ compared to SFT+DPO on AlpacaEval-2, with preference scores provided by GPT-4-Preview-1106 using GPT4-Top1 as reference. **Left**: Absolute bias and variance. **Right**: Absolute error distribution.

challenges. FuseSFT helps regularize the training process, mitigating overfitting to preferences derived from individual source models. Second, FuseSFT enables the target model to generate more diverse responses, which benefits preference optimization in the second stage. In summary, while FuseSFT may initially fall short in delivering superior alignment results, it establishes a more robust foundation that improves preference learning in the second stage. This aligns with our earlier discussion (Section 3.2) and the findings in (Ren & Sutherland, 2025), which show that early supervision signals play a pivotal role in determining the quality of downstream preference optimization.

Furthermore, we observe that following FuseSFT, our proposed FusePO delivers consistently better results compared to existing alignment methods such as DPO. This suggests that FusePO, by effectively balancing the learning signal from diverse multi-source preference pairs, is better equipped to guide the model toward desirable behavior, leading to more robust alignment results.

### 4.4 SCALING WITH THE NUMBER OF SOURCE MODELS

To assess the scalability of FuseRL with respect to the number of source models, we conducted a series of experiments under different configurations. In the single-source setting, we used Gemma2-27B-IT as the only source model. For the two-source configuration, we combined Gemma2-27B-IT with Mistral-Large-Instruct-2407. The four-source setup corresponds to the original FuseRL configuration, incorporating four diverse source models. The results, as summarized in Table 2, reveal a clear and consistent trend: FuseRL achieves progressively stronger alignment performance on AlpacaEval-2 as the number of source models increases from one to four. This underscores the framework's capability to effectively integrate heterogeneous alignment signals and leverage the diversity among source models to improve overall alignment quality.

### 4.5 EFFECT OF FUSERL ON REDUCING BIAS AND VARIANCE

To assess whether FuseRL effectively reduces bias and variance during model fusion, we conducted an experiment using DPO to compare FuseRL with the baseline fusion method (SFT+DPO), which utilizes only one source model per prompt. We analyzed the preference scores (1–2 scale) assigned by GPT-4-Preview-1106 to responses generated by the two fusion methods from AlpacaEval-2. These scores were compared against the preference scores of ideal responses to calculate bias and variance. The goal is to evaluate how the two fusion methods deviate from ideal responses. Since GPT4-Top1 is generated by selecting the top response from source model outputs for each prompt based on GPT-4-Preview-1106, it was used as the reference model to simulate ideal responses.

As shown in Figure 5, FuseRL$_{DPO}$ achieves lower absolute bias and variance compared to SFT+DPO. Specifically, the absolute bias and variance for FuseRL$_{DPO}$ are 0.010 and 0.130, while SFT+DPO shows higher values of 0.021 and 0.135. The absolute error distribution, depicted as box plots, further highlights the advantages of FuseRL$_{DPO}$. The upper whisker of the box plot for FuseRL$_{DPO}$ is lower than SFT+DPO, indicating a tighter and more consistent error distribution. These findings demonstrate that FuseRL reduces bias and variance during the model fusion process. A theoretical analysis of the behavior is provided in Appendix E. We also conducted comparisons using SimPO and RLOO; the corresponding results are included in Appendix K.

Table 3: Comparison of FuseRL and on-policy preference optimization methods (RLOO, SimPO, DPO) on AlpacaEval-2. "SFT" indicates that the target model first perform SFT, followed by on-policy preference optimization.

| Method | AlpacaEval-2 | |
|---|---|---|
| | LC (%) | WR (%) |
| RLOO$_{on}$ | 44.6 | 44.7 |
| SFT+RLOO$_{on}$ | 61.6 | 63.0 |
| FuseRL$_{RLOO}$ | **67.7** (+6.1) | **70.6** (+7.6) |
| SimPO$_{on}$ | 55.3 | 47.2 |
| SFT+SimPO$_{on}$ | 63.0 | 60.5 |
| FuseRL$_{SimPO}$ | **70.6** (+7.6) | **71.3** (+10.8) |
| DPO$_{on}$ | 51.7 | 49.6 |
| SFT+DPO$_{on}$ | 66.3 | 69.8 |
| FuseRL$_{DPO}$ | **70.1** (+3.8) | **70.9** (+1.1) |

Table 4: Comparison of FuseRL and the baseline fusion method (SFT+DPO) on AlpacaEval-2 across different model sizes (1B, 3B, 8B). The results highlight FuseRL's consistent gains in both LC and WR across model scales.

| Size | Method | AlpacaEval-2 | |
|---|---|---|---|
| | | LC (%) | WR (%) |
| 1B | Original | 9.7 | 10.3 |
| | SFT + DPO | 25.6 | 29.7 |
| | FuseRL$_{DPO}$ | **26.8** (+1.2) | **31.0** (+1.3) |
| 3B | Original | 21.4 | 22.6 |
| | SFT + DPO | 47.6 | 50.4 |
| | FuseRL$_{DPO}$ | **50.7** (+3.1) | **57.9** (+7.5) |
| 8B | Original | 28.3 | 28.7 |
| | SFT + DPO | 67.1 | 69.8 |
| | FuseRL$_{DPO}$ | **70.1** (+3.0) | **70.9** (+1.1) |

## 4.6 COMPARISON TO ON-POLICY PREFERENCE OPTIMIZATION

Given that FuseRL leverages preference optimization for model fusion and relies on responses sampled from multiple source models, we conducted experiments to compare it with traditional on-policy preference optimization methods (Rosset et al., 2024; Meng et al., 2024), which use responses sampled exclusively from the target model. To ensure a fairer comparison, we also experimented with the target model to first perform SFT on the best source model for each prompt, followed by self-sampling for preference optimization, using the same training set division as employed in our FuseRL approach. As shown in Table 3, while on-policy methods (RLOO, SimPO, and DPO) outperform direct SFT, their performance still falls short of that achieved by our proposed FuseRL framework. We hypothesize that this gap arises from the lower quality of on-policy responses generated by the target model, which limits the exploration of optimal response spaces, especially when compared to those produced by significantly larger and more capable source models. This limitation explains why performing SFT before preference optimization mitigates the issue and highlights the importance of FuseRL in utilizing high-quality responses from diverse source models.

## 4.7 FUSERL ACROSS MODELS OF DIFFERENT SIZES

To assess the generalizability of FuseRL across different model scales, we conducted additional experiments using Llama-3.2-1B-Instruct and Llama-3.2-3B-Instruct as the target models. These models represent smaller scales compared to the primary 8B model, allowing us to evaluate how well our method performs when applied to models with fewer parameters. The experimental results in Table 4 show that FuseRL consistently achieves higher LC win rates than the baseline method across all scales, including both the 1B and 3B models. This finding demonstrates that FuseRL's ability to fuse heterogeneous source models is not limited to larger target models but also transfers to smaller-scale models. This highlights its potential to enhance alignment across diverse scales.

## 5 CONCLUSIONS

In this paper, we introduced FuseRL to enhance heterogeneous model fusion by maximizing the utilization of multiple source models throughout the alignment process. FuseRL consists of two components: FuseSFT, which integrates the strengths of diverse source models through weighted supervised fine-tuning (SFT) to establish a robust initialization, and FusePO, which optimizes weighted preferences from multiple source outputs to achieve superior alignment. Extensive experiments demonstrate the effectiveness of FuseRL across alignment methods such as RLOO, DPO, and SimPO, and show that it achieves promising performance among 8B-sized LLMs on AlpacaEval-2 and Arena-Hard benchmarks. Our analysis reveals that FuseSFT regularizes the SFT process to prevent overfitting to individual source models and reduce the detrimental squeezing effect, while FusePO introduces diverse preference signals that enhance optimization and alignment with human preferences. These findings highlight FuseRL as a powerful and effective approach for harnessing heterogeneous model knowledge to enhance the optimization and alignment of LLMs.

ETHICS STATEMENT

The primary objective of this work is to enhance model performance by efficiently integrating heterogeneous source models. We believe this approach holds significant potential for improving the alignment of AI systems with human preferences. Throughout the research process, we have adhered to responsible research practices and ethical standards. The datasets used in this study are publicly available and widely recognized within the research community, and we have verified that their use complies with all associated terms and conditions. We confirm that no conflicts of interest or sponsorships have influenced the outcomes of this work.

REPRODUCIBILITY STATEMENT

To support the reproducibility of our work, we have provided comprehensive experimental details in Section 4.1 and Appendix D.3, including data processing procedures, model configurations, and hyperparameter settings. The source code implementing the FuseRL framework is included in the supplementary materials and will be made publicly available.

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

## A    LIMITATIONS

While FuseRL demonstrates strong empirical performance, it also has several limitations. First, the framework relies on an external reward model to assess and weight responses from source models. As a result, the quality of the training signal is sensitive to the alignment and calibration of the reward model. Second, due to resource constraints, the evaluation was conducted on a limited set of source models. The applicability of FuseRL to a broader range of models, including both open-source and commercial LLMs with greater diversity, remains an open direction for future investigation.

## B    STATEMENT ON THE USE OF LARGE LANGUAGE MODELS

In this work, large language models were utilized solely for the purpose of polishing the manuscript. Specifically, they were employed to improve clarity and precision of phrasing, ensure grammatical correctness and spelling accuracy, and provide suggestions to enhance overall coherence and readability. The core research problem, conceptual framework, methodologies, experimental design, analysis, and result interpretation are entirely developed by the authors. The use of LLMs is strictly confined to improving the efficiency and quality of academic writing without influencing the intellectual contributions of this work.

## C    RELATED WORK

This work is closely related to alignment techniques for LLMs, as well as collective approaches like model ensembling and heterogeneous model fusion.

**LLMs alignment**    Aligning large language models (LLMs) with human expectations using techniques such as reinforcement learning from human feedback (RLHF) (Christiano et al., 2017) is a critical step in developing effective and safe LLMs. InstructGPT (Ouyang et al., 2022) employs a three-stage pipeline that includes supervised fine-tuning, reward model training, and policy optimization via proximal policy optimization (PPO) (Schulman et al., 2017). However, this multi-stage process is costly, complex, and potentially unstable. To address these challenges, researchers have explored various improvements. For instance, Ahmadian et al. (2024) showed that simplified reinforcement learning methods such as REINFORCE (Williams, 1992) can achieve alignment effectively without relying on advanced optimization components like value-function critics and advantage estimation. Similarly, reinforcement learning from AI feedback (RLAIF) (Lee et al., 2024) offers a cost-effective alternative to relying on expensive human-labeled data by utilizing preference labels generated by LLMs, while achieving comparable performance to traditional RLHF methods.

Direct Preference Optimization (DPO) (Rafailov et al., 2023) simplifies the RLHF process by directly optimizing the policy using human preference data, eliminating the need for an explicit reward model and offering improved training stability. However, DPO faces challenges, such as its reliance on a reference model, susceptibility to overfitting on noisy preference data, and managing the trade-off between exploration and exploitation. ORPO (Hong et al., 2024) addresses the dependency of DPO on a reference model by incorporating odds ratios into the supervised fine-tuning process, allowing models to directly distinguish between preferred and dispreferred outputs. KTO (Ethayarajh et al., 2024) introduces a human-aware loss (HALO) to maximize the utility of model generations using a binary signal indicating desirability, rather than focusing on preference likelihoods. Similarly, RSO (Liu et al., 2024) enhances preference optimization by sourcing data pairs from the estimated optimal policy through rejection sampling. Recently, SimPO (Meng et al., 2024) further streamlines DPO by leveraging the average log-probability of sequences as an implicit reward and introducing a reward margin to better differentiate between positive and negative responses.

**Collective LLMs**    Collective LLMs aim to enhance the performance of LLMs by integrating knowledge and capabilities from multiple models. As a representative ensemble method, LLM-Blender (Jiang et al., 2023b) performs pairwise ranking of candidate outputs, selecting and aggregating the most promising responses into a superior output using a sequence-to-sequence model. Similarly, Mixture-of-Agents (MoA) (Wang et al., 2024b) employs a multi-layer architecture, where LLM agents in each layer iteratively refine responses based on the outputs of the previous layer, gradually improving generation quality. UltraFuser (Ding et al., 2024) leverages three expert models trained on language, code, and mathematics tasks, and combines their outputs through a token-level gating mechanism to dynamically select the most relevant expertise for each task. Branch-Train-MiX (BTX) (Sukhbaatar et al., 2024) employs a parallel training strategy to train multiple expert

models starting from a shared seed model, which are combined into a Mixture of Experts (MoE) framework. The resulting MoE model is then fine-tuned to optimize token-level routing decisions and maximize the utilization of each expert's capabilities.

Heterogeneous model fusion aims to transfer the capabilities of multiple source models into a single target model. These approaches can be broadly classified as explicit or implicit. Explicit model fusion (EMF) methods, such as FuseLLM (Wan et al., 2024a) and FuseChat (Wan et al., 2024b), utilize knowledge distillation to explicitly transfer knowledge, typically in the form of probabilistic distribution matrices, from multiple source models to a single target model. FuseLLM employs a multi-teacher distillation strategy for this transfer, whereas FuseChat adopts a fuse-and-merge framework. In FuseChat, pairwise knowledge fusion is first conducted between each source model and a pivot model to produce multiple target models with identical structure and size. These target models are then merged within the parameter space to complete the process. WRPO (Yang et al., 2024c) introduces implicit model fusion (IMF), where the target model leverages high-quality responses generated by source models as auxiliary signals during preference optimization. However, WRPO focuses solely on selecting the highest-reward output for each prompt, which limits the utilization of the broader knowledge from all source models. This neglect of the diverse and rich signals from source LLMs may limit the effectiveness of model fusion.

## D  IMPLEMENTATION DETAILS

### D.1  DETAILS OF PRELIMINARY EXPERIMENTS

In this section, we provide a detailed description of the experimental setup used in our preliminary experiments, with the results illustrated in Figure 1. The data construction process for these preliminary experiments mirrors that of the main experiment described in Section 4.1, utilizing the same four source models and reward model. For each prompt in the UltraFeedback test set (Cui et al., 2024), each source model generates five responses, which are then scored by the reward model, ArmoRM-Llama3-8B-v0.1 (Wang et al., 2024a). We compare our proposed method, FuseRL, against SFT+PO, which serves as a baseline implementation of our approach. Specifically, SFT+PO incorporates only a single response during supervised fine-tuning (SFT) or a single preference pair during preference optimization (PO) for each prompt. In this context, we explore preference optimization using a range of techniques, including RLOO (Ahmadian et al., 2024), SimPO (Meng et al., 2024), and DPO (Rafailov et al., 2023).

To evaluate the impact of FuseRL on the model's ability to distinguish response quality, we conduct two types of evaluations: Intra-Rank and Cross-Rank. The Intra-Rank evaluation examines the model's ability to distinguish response quality within a *single* source model, while the Cross-Rank evaluation assesses its ability to distinguish response quality across *different* source models. In the **Intra-Rank** evaluation, for each source model, the reward model identifies the response with the highest reward $y_w$ and the one with the lowest reward $y_l$. Following previous study (Meng et al., 2024), the model under evaluation computes the average log probability for each response as its predicted reward score $r_m(y)$. It is important to note that for DPO and RLOO, the computation of rewards during evaluation differs from their training phase but remains consistent with their inference phase. To ensure fairness, we adopt the same approach described above for all three methods: RLOO, SimPO, and DPO. We then check whether $r_m(y_w) > r_m(y_l)$ and calculate the accuracy as the ratio of correct matches to the total number of samples in the test set for each source model. The final result is obtained by averaging the accuracy across all source models. As for the **Cross-Rank** evaluation, we select one response from each source model for each test prompt. The reward model then identifies the response with the highest reward $y_w$ and the response with the lowest reward $y_l$. We verify whether $r_m(y_w) > r_m(y_l)$ following the same process as the Intra-Rank evaluation and calculate the accuracy as the ratio of correct matches to the total number of samples in the test set.

### D.2  DETAILS OF BASELINES

We evaluate our method against various baseline models: proprietary LLMs, source and target LLMs, ensemble LLMs, and heterogeneous model fusion approaches.

**Proprietary LLMs**: We evaluate closed-source models, including GPT-4o (OpenAI, 2024), GPT-4-Turbo (Achiam et al., 2023). We prioritize results from official sources.

**Source and Target LLMs**: The evaluation strategy mirrors that used for Proprietary LLMs, relying on official results when available and locally evaluated results otherwise.

**Ensemble LLMs**: Ensemble LLMs leverage multiple models to enhance performance through various collaborative approaches. In this study, we examine several methods for utilizing responses from our source LLMs. The GPT4-Top1 (Achiam et al., 2023) provides an upper performance bound by ranking the responses from source models based on GPT-4's evaluations and selecting the best one. Similarly, LLM-Blender-Top1 (Jiang et al., 2023b) employs a ranking mechanism to choose the optimal response from multiple LLM outputs. Alternatively, the MoA (Wang et al., 2024b) uses Qwen2.5-72B-Instruct as an aggregator to integrate responses and produce a unified output.

**Heterogeneous Model Fusion.** FuseLLM (Wan et al., 2024a) and FuseChat (Wan et al., 2024b) adopt knowledge distillation techniques to transfer knowledge from multiple source models to a target model. Due to computational constraints, we did not reproduce these results using our specific source and target models. Instead, we rely on the results reported by Yang et al. (2024c), while noting minor differences in the number and versions of the source models used. Furthermore, we compare our approach with WRPO (Yang et al., 2024c), the work most closely related to ours.

### D.3 DETAILS OF HYPERPARAMETERS

All our experiments were conducted using the TRL (von Werra et al., 2020) library. The Ultra-Feedback (Cui et al., 2024) dataset was randomly divided into two subsets in a 4:6 ratio for the two-stage training process. For on-policy implementation, all samples were directly used for training. A batch size of 128 and a maximum sequence length of 2048 were applied across all stages.

During the SFT/FuseSFT stage, training was performed over 3 epochs. The learning rate was selected through a search over the range [1e-6, 7e-6, 1e-5, 2e-5], with 7e-6 chosen for SFT and 1e-5 for FuseSFT. For the FuseSFT/FusePO stage, the temperature parameter was explored within the range [1e-1, 1e-2, 5e-3, 1e-3, 1e-4], with 1e-2 chosen for FuseSFT, 5e-3 for FuseRL$_{DPO}$ and FuseRL$_{RLOO}$, and 1e-3 for FuseRL$_{SimPO}$. For the implementation of RLOO in the TRL library, a KL penalty is essential to prevent training collapse. The KL coefficient was selected from the range [1e-4, 1e-3, 1e-2, 1e-1]. In the preference optimization stage, the search strategy from SimPO (Meng et al., 2024) was followed. The learning rate search range for all preference learning algorithms was [3e-7, 5e-7, 6e-7, 8e-7, 1e-6].

The best hyperparameter settings for some baselines and FuseRL are summarized in Table 5. For response collection, we utilized the vLLM library (Kwon et al., 2023). The sampling parameters for each source model were configured based on their default generation settings. Detailed sampling parameters for the various source models are provided in Table 6. All experiments were conducted on a computing cluster equipped with 8x80G NVIDIA A800 GPUs.

Table 5: Hyperparameter configurations for various approaches in the main experiment, where $\alpha$ represents the weight used in the progressive learning strategy of WRPO, and "KL Coef." denotes the KL coefficient applied in RLOO.

| Method | $\beta$ | $\gamma$ | $\alpha$ | KL Coeff. | Learning Rate |
|---|---|---|---|---|---|
| RLOO | - | - | - | 1e-2 | 5e-7 |
| SimPO | 10.0 | 3 | - | - | 6e-7 |
| DPO | 1e-2 | - | - | - | 3e-7 |
| SFT + RLOO | - | - | - | 1e-2 | 1e-6 |
| SFT + SimPO | 10.0 | 3 | - | - | 1e-6 |
| SFT + DPO | 1e-2 | - | - | - | 1e-6 |
| SFT + WRPO | 1e-2 | - | 1e-1 | - | 1e-6 |
| FuseRL$_{RLOO}$ | - | - | - | 1e-2 | 1e-6 |
| FuseRL$_{SimPO}$ | 10.0 | 3 | - | - | 1e-6 |
| FuseRL$_{DPO}$ | 1e-2 | - | - | - | 1e-6 |

Table 6: Sampling parameters for different models

| Model | $p$ | Temperature | Repetition penalty |
|---|---|---|---|
| Llama-3.1-8B-Instruct | 0.8 | 0.6 | 1.0 |
| Mistral-Large-Instruct | 0.95 | 0.8 | 1.0 |
| Gemma2-27B-IT | 0.95 | 0.8 | 1.0 |
| Qwen2.5-72B-Instruct | 0.8 | 0.7 | 1.05 |
| DeepSeek-V2-Chat | 0.95 | 0.8 | 1.0 |

## E THEORETICAL ANALYSIS

We conduct a theoretical analysis of FuseSFT and FusePO to illustrate how reward-based weighting aggregation enhances the robustness and effectiveness of the FuseRL framework.

**Proposition 1.** *In both FuseSFT and FusePO, reward-based weighting aggregation emphasizes responses with higher weights during parameter updates, progressively guiding the target model to prioritize high-quality responses throughout fine-tuning and preference optimization.*

*Proof.* Consider the loss functions defined in Eq. (10) and Eq. (11). In both cases, the loss is represented as a weighted sum of model-specific losses, with the weights determined by $w_{x,i}$. For ease of analysis, we introduce a unified notation $\mathcal{L}$, which represents both $\mathcal{L}_{\text{sft}}$ and $\mathcal{L}_{\text{pref}}$. Moreover, we use $\mathcal{L}_i$ to denote the model-specific loss component within the two loss functions. Therefore, the gradient of the loss $\mathcal{L}$ with respect to model parameters $\theta_T$ is given by:

$$\nabla_{\theta_T}\mathcal{L} = \sum_{x \in \mathcal{X}} \sum_{i=1}^{K} w_{x,i} \cdot \nabla_{\theta_T}\mathcal{L}_i. \tag{12}$$

Since $w_{x,i}$ is derived from a softmax function, source models with higher maximum rewards are assigned exponentially larger weights. This amplifies the scaling of their corresponding gradients, $\nabla_{\theta_T}\mathcal{L}_i$, by larger factors. As a result, these high-weighted terms dominate the overall gradient, steering parameter updates to prioritize minimizing the loss associated with high-reward responses. Furthermore, we note that the parameter updates also depend on the magnitude of $\nabla_{\theta_T}\mathcal{L}_i$. If a highly weighted term has a small gradient, its influence on the parameter updates may be limited. Nevertheless, the weighted aggregation naturally amplifies the relative importance of high-reward terms, ensuring they receive greater attention during optimization.

**Proposition 2.** *Under the assumptions that the biases introduced by different source models are independent and identically distributed (i.i.d.) for each input $x \in \mathcal{X}$, aggregating and weighting responses or preference pairs from multiple source models preserves the expected bias of individual models and strictly reduces their variance.*

*Proof.* Let $\epsilon_{x,i}$ represent the bias introduced by source model $M_i$ for a given input $x \in \mathcal{X}$. The aggregated influence of these biases on the gradient update is:

$$\epsilon_{\text{agg}}(x) = \sum_{i=1}^{K} w_{x,i} \cdot \epsilon_{x,i}. \tag{13}$$

Since these biases are independent and identically distributed, it follows that $\mathbf{E}[\epsilon_{x,i}] = \mu$ and $\text{Var}(\epsilon_{x,i}) = \sigma^2$.

The expected value of the aggregated bias is the sum of the expected values of each weighted bias:

$$\mathbb{E}[\epsilon_{\text{agg}}(x)] = \mathbb{E}\left[\sum_{i=1}^{K} w_{x,i} \cdot \epsilon_{x,i}\right]$$
$$= \sum_{i=1}^{K} w_{x,i} \cdot \mathbb{E}[\epsilon_{x,i}] = \mu \sum_{i=1}^{K} w_{x,i} = \mu. \tag{14}$$

The variance of the aggregated bias is given by:

$$\text{Var}\left(\epsilon_{\text{agg}}(x)\right) = \text{Var}\left(\sum_{i=1}^{K} w_{x,i} \cdot \epsilon_{x,i}\right) = \sum_{i=1}^{K} w_{x,i}^2 \cdot \text{Var}(\epsilon_{x,i}). \tag{15}$$

Since $w_{x,i}$ are weights derived from softmax normalization, we have $0 < w_{x,i} < 1$ and $\sum_{i=1}^{K} w_{x,i} = 1$. Therefore, $w_{x,i}^2 < w_{x,i}$, and summing over all $i$ yields:

$$\sum_{i=1}^{K} w_{x,i}^2 < \sum_{i=1}^{K} w_{x,i} = 1. \tag{16}$$

Thus, by combining Equations (15) and (16), we obtain:

$$\text{Var}\left(\sum_{i=1}^{K} w_{x,i} \cdot \epsilon_{x,i}\right) < \text{Var}(\epsilon_{x,i}) = \sigma^2. \tag{17}$$

For every input $x$, the expected value of the aggregated bias $\epsilon_{\text{agg}}(x)$ remains equal to the expectation of the individual biases, $\mu$, ensuring that the aggregation process preserves the systematic bias. Moreover, the variance of the aggregated bias is strictly less than $\sigma^2$, demonstrating that aggregating and weighting the biases effectively reduces variance.

Table 7: Evaluation results of FuseRL on various downstream tasks.

| Dataset (→) Setup (→) Metric (→) | HellaSwag 10-shot Acc Norm | MuSR 0-shot Acc Norm | MMLU-Pro 5-shot Acc | GPQA Diamond 0-shot Acc Norm | SciQ 0-shot Acc Norm | MMLU-Redux 0-shot Acc | AMC 23 0-shot, CoT Acc | LiveCodeBench (2408-2411) 0-shot Pass @ 1 | Avg. |
|---|---|---|---|---|---|---|---|---|---|
| Llama-3.1-8B-Instruct | 80.2 | 35.7 | 33.6 | 33.8 | 96.0 | 67.2 | 25.0 | 12.3 | 53.1 |
| SFT | 62.3 | 38.8 | 36.7 | 31.8 | 92.4 | 68.6 | 27.5 | 11.3 | 51.2 |
| FuseSFT | 80.8 | 39.4 | 35.2 | 31.3 | 96.3 | 65.0 | 17.5 | 10.0 | 52.2 |
| SFT + DPO | 83.6 | 34.7 | 37.1 | 29.3 | 87.1 | 68.4 | 25.0 | 11.3 | 52.2 |
| SFT + WRPO | 84.1 | 33.7 | 36.5 | 28.8 | 94.6 | 66.3 | 17.5 | 9.4 | 51.6 |
| FuseRL$_{DPO}$ | 82.0 | 34.9 | 34.7 | 33.3 | 95.7 | 66.5 | 27.5 | 12.5 | 53.5 |

# F  DOWNSTREAM TASK EVALUATION

To assess FuseRL's impact on downstream tasks, we conducted experiments on eight downstream tasks spanning general knowledge, mathematics, and coding. These tasks are described as follows:

**HellaSwag** (Zellers et al., 2019): A commonsense reasoning benchmark requiring models to choose the most plausible continuation of a given context.

**MuSR** (Sprague et al., 2024): A dataset comprising algorithmically generated complex problems, such as murder mysteries, object placement challenges, and team allocation optimizations. These tasks require advanced reasoning skills and the ability to parse long-range context effectively.

**MMLU-Pro** (Wang et al., 2024c): An enhanced version of MMLU (Hendrycks et al., 2021), which is a multiple-choice dataset to evaluate knowledge capability. This dataset is designed to address issues such as noisy data and reduced difficulty due to advances in model capabilities and increased data contamination. MMLU-Pro increases challenge levels by expanding multiple-choice options from 4 to 10, requiring reasoning across more questions, and incorporating expert-reviewed annotations for improved quality and reduced noise.

**GPQA Diamond** (Rein et al., 2023): A challenging knowledge benchmark crafted by PhD-level domain experts in biology, physics, and chemistry. The dataset contains questions that are straightforward for experts but difficult for laypersons. We evaluate on the highest quality diamond set comprising 198 questions.

**SciQ** (Welbl et al., 2017): A collection of 13.7k multiple-choice questions derived from science exams, covering a broad range of scientific topics.

**MMLU-Redux** (Gema et al., 2024): A re-annotated subset of the MMLU (Hendrycks et al., 2021) dataset created through manual assessment from 14 human experts.

**AMC 23** (Yang et al., 2024a): The 2023 American Mathematics Competition, featuring 25 multiple-choice questions that test advanced high school mathematics, including trigonometry, advanced algebra, and elements of calculus.

**LiveCodeBench (2408-2411)** (Jain et al., 2024): A benchmark designed to evaluate coding capabilities using an evolving set of contamination-free problems sourced from platforms including LeetCode, AtCoder, and CodeForces. We evaluate on the subset comprising 160 problems published between August 2024 and November 2024.

The results presented in Table 7 offer several important insights. Both SFT and FuseSFT lead to a decline in general performance. This decrease can be attributed to the fact that our training dataset primarily emphasizes preference alignment, suggesting an inherent trade-off between preference alignment and overall model performance. Although FuseSFT does not surpass SFT in alignment performance, it performs better at preserving the model's general capabilities. This highlights FuseSFT's strength in balancing human preference alignment while maintaining broader performance. After the preference alignment stage, a slight improvement in general performance is observed across the models. However, with the exception of FuseRL, all models perform worse than the target model. Interestingly, the average performance of FuseRL exceeds that of the target model, albeit by a small margin. This indicates that FuseRL not only improves preference alignment but also effectively maintains general performance.

## G    TRAINING COST ANALYSIS FOR MODEL FUSION

FuseRL is designed with a scalable data strategy that enables efficient use of training resources while maintaining strong performance. In particular, our framework supports data scaling along two complementary dimensions: the number of prompts and the number of responses per prompt. This dual-scaling mechanism allows the model to benefit from a richer distribution of supervision signals without proportionally increasing the cost of data preparation.

Notably, scaling the number of responses is relatively efficient—it only requires sampling from different source models. In contrast, scaling the number of prompts involves a more complex pipeline that includes classification, filtering, and rewriting, which is significantly more resource-intensive.

Despite this, FuseRL maintains strong alignment performance under constrained training budgets. As shown in Table 8, using only 15K prompts with 4 responses per prompt, our method matches the performance of baselines trained with 60K prompts and a single response. Moreover, while these baselines exhibit signs of performance saturation, FuseRL continues to benefit from larger datasets. When scaled to 60K prompts with 4 responses, FuseRL yields further improvements, highlighting its superior scaling potential. All experiments were conducted on a cluster of $8 \times 80\text{GB}$ A800 GPUs.

Table 8: FuseRL achieves competitive or superior performance with fewer prompts and more responses, demonstrating better scalability compared to baseline methods.

| Method | Prompts | Responses | Runtime (hrs) | AlpacaEval-2 | |
|---|---|---|---|---|---|
| | | | | LC (%) | WR (%) |
| FuseRL$_{\text{DPO}}$ | 60K | 4 | 11.5 | **70.1** | 70.9 |
| | 30K | 4 | 6.2 | 69.0 | 72.5 |
| | 15K | 4 | 3.8 | 64.2 | 69.2 |
| SFT+DPO | 60K | 1 | 3.6 | 67.1 | 69.2 |
| SFT+WRPO | 60K | 1 | 4.8 | 67.7 | **74.2** |

## H    IMPACT OF DIFFERENT $k$ ON FUSERL

In this section, we examine the impact of varying the number of responses and preference pairs, denoted as $k$ (where $1 \le k \le K$), on the final alignment performance of the target model in both stages of FuseSFT and FusePO. Specifically, $k$ in the FuseSFT stage refers to the top-$k$ responses from all source models used in Eq. (10), ranked by reward scores, while in the FusePO phase, it represents preference pairs derived from the top-$k$ highest-scoring source models used in Eq. (11). These variations in the selection of responses and preference pairs are evaluated to understand their influence on the alignment performance of the target model. As shown in Figure 6, increasing $k$ in both the FuseSFT and FusePO stages leads to consistent performance improvement in the target model's LC win rate. This indicates that our method effectively leverages responses (even suboptimal responses and preference pairs) from multiple source models for optimization.

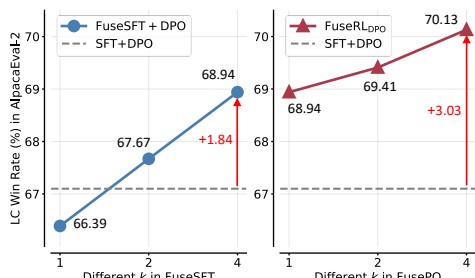

Figure 6: The impact of varying the number of responses or preference pairs of FuseSFT and FusePO on the LC win rate. **Left**: Results for FuseSFT + DPO, where $k$ denotes using the top-$k$ responses from source models during the FuseSFT stage. **Right**: Results for FuseRL$_{\text{DPO}}$, denoting using preference pairs derived from the top-$k$ highest-rewarding source models for the FusePO.

## I    TEMPERATURE COEFFICIENTS IN FUSERL

The temperature coefficient play a crucial role in weighting the contributions of responses or preference pairs from different source models, calculated using a softmax-based reward mechanism as defined in Eq. (8). In this section, we examine the influence of different temperature

coefficients on the performance of the FuseRL framework, which consists of two stages: FuseSFT and FusePO, with SFT+DPO serving as the baseline. The effect of temperature coefficients in the FuseSFT stage is demonstrated through the results of FuseSFT followed by off-policy DPO training. For the FusePO stage, we use the optimal settings identified for FuseSFT and analyze the influence of adjusting the temperature parameter on FusePO performance.

In Figure 7, we observe consistent performance improvements of FuseSFT and FusePO compared to the SFT+DPO baseline across a wide range of temperature settings. This clearly demonstrates the effectiveness of the reward-based weighting mechanism in integrating diverse information from heterogenes source models, enabling the target model to achieve superior performance.

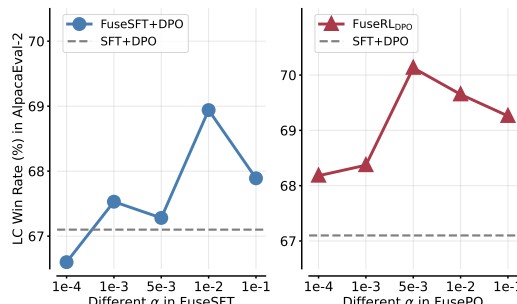

Figure 7: The influence of varying temperature coefficients $\alpha$ on the performance of FuseRL, including FuseSFT and FusePO stages, on AlpacaEval-2.

## J    RESPONSES SELECTION STRATEGIES FOR FUSESFT

In this section, we analyze the impact of various response selection strategies on the performance of FuseSFT, focusing on how different methods influence the model's alignment performance. To illustrate these effects, we present the results of FuseSFT trained with different strategies, along with the outcomes of subsequent DPO training. The first strategy, which serves as the default configuration, the top-$k$ responses from all available responses generated by the source models. In this case, $k = 4$, meaning the top four responses across all source models are chosen. The second strategy selects the top response from each source model, resulting in a total of four responses (one per source model). The results in Table 9 demonstrate a clear hierarchy: the top-$k$ selection strategy outperforms the top-1 selection per source model, regardless of the training stage. These findings highlight the critical importance of prioritizing high-quality responses during the alignment process. The top-$k$ selection strategy not only leverages the advantage of weighted responses from multiple source models but also consistently delivers the best results by utilizing the most informative and relevant responses.

Table 9: Comparison of different response selection strategies for FuseSFT on AlpacaEval-2.

| Method | Settings | AlpacaEval-2 | |
|---|---|---|---|
| | | LC (%) | WR (%) |
| FuseSFT | Top-$k$ from all source models | **38.8** | **33.7** |
| | Top-1 from each source models | 36.3 | 31.6 |
| FuseSFT+DPO | Top-$k$ from all source models | **68.9** | **73.0** |
| | Top-1 from each source models | 66.5 | 71.2 |
| FuseRL$_{DPO}$ | Top-$k$ from all source models | **70.1** | **70.9** |
| | Top-1 from each source models | 68.7 | 70.2 |

## K    SUPPLEMENTARY ANALYSIS OF FUSERL: REDUCING BIAS AND VARIANCE

In Section 4.5, we analyze the impact of FuseRL on reducing bias and variance by conducting analytical experiments. These experiments compare the responses generated by different approaches with the simulated ideal responses (by GPT4-Top1) on AlpacaEval-2. Below, we first detail the evaluation metrics, including absolute error, absolute bias, and variance:

- **Absolute Error**: The absolute difference between the preference scores of the response generated by the model under study and the response by GPT4-Top1.

- **Absolute Bias**: The mean of the absolute errors across all data points.

- **Variance**: The mean squared deviation of the absolute errors, indicating the consistency of the model's predictions.

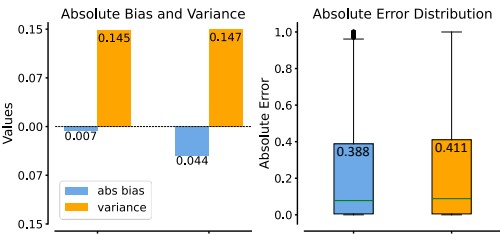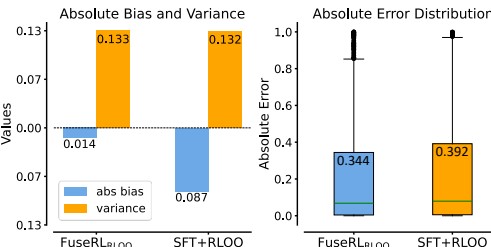

Figure 8: Comparison of absolute bias, variance, and absolute error distribution between FuseRL and baseline methods. **Left**: FuseRL$_{SimPO}$ vs. SFT+SimPO. **Right**: FuseRL$_{RLOO}$ vs. SFT+RLOO.

Furthermore, we present supplementary experimental results to further support our findings. In Figure 8 (Left), we compare FuseRL$_{SimPO}$ with the baseline, while in Figure 8 (Right), we compare FuseRL$_{RLOO}$ with SFT+RLOO.

These supplementary results demonstrate that FuseRL achieves measurable reductions in absolute bias compared to relying solely on the best individual source model for each prompt, highlighting its effectiveness in minimizing deviations between the generated and (simulated) ideal responses. Moreover, FuseRL (except for RLOO) demonstrates lower variance, indicating enhanced consistency and robustness in generating responses aligned with human preferences. However, while RLOO under the FuseRL framework achieves a substantial reduction in bias, its variance shows a slight increase. This can be attributed to two factors. First, due to computational resource limitations, RLOO uses only two responses per prompt, which restricts its overall performance and affects the variance scores. Second, there is an inherent trade-off between bias and variance—RLOO's optimization strategy prioritizes minimizing bias, which increases sensitivity to input variations and leads to a slight rise in variance. Moreover, the absolute error distributions under FuseRL are consistently lower than those of the baseline methods, further emphasizing its ability to deliver stable and consistent performance across diverse inputs.

