# OpenReview forum: "FuseRL: Dense Preference Optimization for Heterogeneous Model Fusion"
_ICLR.cc/2026/Conference — ICLR 2026 Conference Withdrawn Submission_

### Official Review · Reviewer_uJzB · 2025-10-25

**Soundness:** 3
**Presentation:** 3
**Contribution:** 3
**Rating:** 6
**Confidence:** 3

**Summary:**

This paper introduces FuseRL, a novel two-stage framework for heterogeneous model fusion. The method aims to address the limitation of existing approaches that often rely on a single "best" output per prompt from source models, which leads to sparse training signals and potential overfitting. The authors demonstrate the effectiveness of their approach by integrating four large, heterogeneous source models into a smaller (8B) target model (Llama-3.1-8B-Instruct), achieving state-of-the-art performance among 8B models on the AlpacaEval-2 and Arena-Hard benchmarks. The framework is shown to be compatible with various preference optimization methods (RLOO, DPO, SimPO).

**Strengths:**

The core idea of moving beyond a single best response per prompt to "dense" utilization of multiple source models is timely, well-argued, and addresses a clear gap in the heterogeneous model fusion literature. The concept of using reward-based weighting to aggregate signals from diverse models is intuitive and novel.

The results are impressive. Achieving performance that competes with or surpasses much larger source models and proprietary APIs (like GPT-4-Turbo) using an 8B parameter model is a significant result. The consistent improvements over strong baselines (SFT+PO, WRPO) across multiple preference learning algorithms robustly validates the proposed method's effectiveness.

The paper is thorough in its evaluation. Using four large, architecturally diverse source models and a smaller target model provides a compelling and challenging testbed. The choice of benchmarks (AlpacaEval-2, Arena-Hard) is standard and appropriate.

The paper is generally well-written, with clear explanations, helpful figures, and a well-structured narrative. The appendices are extensive and provide necessary details for reproducibility.

**Weaknesses:**

While the empirical results are strong, the theoretical analysis in Appendix E, while a positive addition, feels somewhat secondary. The propositions are relatively straightforward consequences of the weighted aggregation scheme. A deeper theoretical insight into why this dense signal leads to better generalization and alignment would strengthen the contribution. The connection to the mentioned "squeezing effect" could also be explored more formally.

The source models, while diverse in architecture and scale, are all high-quality, instruction-tuned models. It is unclear how the method would perform if the source model pool included lower-quality or misaligned models. The weighting scheme should theoretically down-weight them, but this edge case is not explored.

Although Section 4.5 and Appendix G discuss training cost, the upfront cost of generating multiple responses from several large source models for the entire training dataset is substantial and could be a barrier for some research groups. A more explicit discussion of this trade-off (performance vs. data generation cost) in the main text would be beneficial.

**Questions:**

How sensitive is FuseRL to the choice of the external reward model? Have you experimented with different RMs, and if so, how does the performance of the final fused model vary?

The weighting scheme in Eq. (8) uses the maximum reward from a model's sampled responses. Did you explore other aggregation methods for a model's set of responses (e.g., average reward) to compute its weight ?

The results on downstream tasks show that FuseRL is better at preserving general capabilities compared to baselines, but the gains are modest. Can you speculate on how to further improve the framework to achieve more significant gains on general benchmarks while excelling in alignment?

---

### Official Review · Reviewer_xP1E · 2025-10-31

**Soundness:** 3
**Presentation:** 3
**Contribution:** 2
**Rating:** 4
**Confidence:** 4

**Summary:**

This paper presents a new, two-step method called FuseRL for combining different AI language models to create a stronger one. Instead of the common approach of just picking the single best answer from a group of AIs, FuseRL uses all of their outputs to learn more effectively.
In the first step, it uses the top-rated answers from several models to give a new AI a strong starting point. In the second step, it fine-tunes the new AI by having it learn from pairs of good and bad answers from the entire pool of responses. The authors show that their medium-sized AI, created with FuseRL, can perform on par with—and sometimes even outperform—models that are much larger.

**Strengths:**

1. The two-stage FuseSFT and FusePO design is intuitive. Using a weighted SFT to create a strong initial model and then leveraging complete response sets for preference optimization is an intuitive way to balance diversity and quality. The framework is clearly explained and visualized.
2. The authors conduct extensive experiments.

**Weaknesses:**

While the paper demonstrates positive empirical results, the contribution appears to be an incremental extension of existing work. The framework combines well-established techniques in a logical but straightforward manner. As such, the work may lack the significant conceptual novelty and technical depth expected for a publication at ICLR.

Few notes:

1. The i.i.d. assumption in Proposition 2 is fundamentally at odds with the paper's heterogeneous setting. This invalidates the formal proof regarding bias and variance reduction.
2. The claim of universal improvement is contradicted by the results in Table 1, where FuseRLDPO underperforms the SFT+DPO baseline on the raw Win Rate for Arena-Hard (57.5% vs. 57.6%). This exception needs to be addressed in the text.

**Questions:**

None

---

### Official Review · Reviewer_4j3B · 2025-11-01

**Soundness:** 3
**Presentation:** 3
**Contribution:** 2
**Rating:** 2
**Confidence:** 3

**Summary:**

This paper presents FuseRL, a two-stage framework designed to fuse heterogeneous LLMs by leveraging both SFT and preference optimization. Specifically, it consists of:
(1) FuseSFT, which integrates responses from multiple source LLMs through weighted supervised fine-tuning; and
(2) FusePO, which further optimizes the target model using dense preference signals aggregated from all sources.
The approach is evaluated across several offline preference optimization paradigms (RLOO, DPO, SimPO) and multiple benchmarks such as AlpacaEval-2 and Arena-Hard.

**Strengths:**

1. The paper is well-organized, with clear motivation, coherent figures, and consistent equations that effectively support the overall argument.
2. The proposed FuseRL systematically integrates supervised fine-tuning (FuseSFT) and preference optimization (FusePO), enabling effective fusion of heterogeneous LLMs to maximize the utility of diverse model outputs.
3. Extensive experiments across AlpacaEval-2, Arena-Hard, and multiple alignment methods (RLOO, DPO, SimPO) demonstrate significant improvements over current baselines. Moreover, the ablations further confirm consistent gains as more sources are involved.

**Weaknesses:**

1. My main concern is the lack of novelty. The method is largely an extension of existing data distillation or response aggregation techniques rather than a fundamentally new fusion paradigm. Distilling responses from multiple models into a single model is relatively straightforward and has been explored in multimodal domains (e.g., SILKIE [1]).
2. The paper observes a decrease in response length compared with standard DPO/SimPO but does not analyze the cause. It remains unclear whether this is a byproduct of the fusion procedure.
3. Large-scale data collection is costly. What about the computational costs in the data collection procedure?

I would be willing to reconsider my rating if the authors can provide convincing justification or additional analysis.
[1] SILKIE: PREFERENCE DISTILLATION FOR LARGE VISUAL LANGUAGE MODELS, 2023.

**Questions:**

My question lies in the reasoning performance evaluation.
Has FuseRL been tested on reasoning-centric benchmarks such as AIME24 or AIME25?
If so, do the performance gains from multi-source fusion also extend to reasoning and mathematical tasks, where compositional and chain-of-thought behaviors are more critical?

---

### Official Review · Reviewer_zpoT · 2025-11-01

**Soundness:** 2
**Presentation:** 2
**Contribution:** 1
**Rating:** 2
**Confidence:** 5

**Summary:**

FuseRL is a two-stage heterogeneous model fusion framework. FuseSFT performs weighted SFT over responses from multiple source LLMs; FusePO performs weighted preference optimization (RLOO/DPO/SimPO) using multi-source response sets.

**Strengths:**

1. Ablations separating FuseSFT vs. FusePO and bias/variance analysis provide some insight into why it helps.

2. Data construction, weighting, and two-stage optimization are described at a level that an experienced practitioner could reimplement.

**Weaknesses:**

1. In Table 1, WRPO has the best AlpacaEval-2 WR (74.1) while FuseRL’s best WR is 71.3, yet the abstract claims SoTA among 8B models “on AlpacaEval-2 and Arena-Hard.” If SoTA is only for LC (length-controlled) or Arena-Hard WR/SC, say so explicitly and harmonize claims in abstract, intro, and results.

2. Both benchmarks are judged by GPT-4-1106, which may bias toward specific styles. No human evals, multiple judges, or cross-judge robustness are reported.

3. Proposition 1 is almost tautological. Proposition 2 treats weights as constants independent of biases and assumes i.i.d. biases across sources; with softmax weights derived from the same rewards that reflect those biases, independence is doubtful. The strict “variance < σ²” claim also fails when a single weight ≈1.

4. The same external RM (ArmoRM-Llama3-8B) controls both selection and weighting; the target is also Llama-3.x-8B. This can bias toward responses that the RM favors stylistically and may inflate gains when judged by GPT-4.

5. Selecting the top-4 responses across all sources for FuseSFT and using max/min per source for FusePO are heuristic. The paper claims within-source pairs reduce “distributional variance,” but cross-source pairs might increase useful diversity.

6. No details on FLOPs, wall-clock, GPUs, batch sizes, sequence lengths, or training stability (divergence rates).

**Questions:**

See weakness part

---

### Note · Authors · 2025-12-29

I have read and agree with the venue's withdrawal policy on behalf of myself and my co-authors.